# Robust Automatic Modulation Classification with Fuzzy Regularization

Xinyan Liang [1]   Ruijie Sang [1]   Yuhua Qian [1]   Qian Guo [2]   Feijiang Li [1]   Liang Du [1]

## Abstract

Automatic modulation classification (AMC) serves as a foundational pillar for cognitive radio systems, enabling critical functionalities including dynamic spectrum allocation, non-cooperative signal surveillance, and adaptive waveform optimization. However, practical deployment of AMC faces a fundamental challenge: prediction ambiguity arising from intrinsic similarity among modulation schemes and exacerbated under low signal-to-noise ratio (SNR) conditions. This phenomenon manifests as near-identical probability distributions across confusable modulation types, significantly degrading classification reliability. To address this, we propose Fuzzy Regularization-enhanced AMC (FR-AMC), a novel framework that integrates uncertainty quantification into the classification pipeline. The proposed FR has three features: (1) Explicitly model prediction ambiguity during backpropagation, (2) dynamic sample reweighting through adaptive loss scaling, (3) encourage margin maximization between confusable modulation clusters. Experimental results on benchmark datasets demonstrate that the FR achieves superior classification accuracy and robustness compared to compared methods, making it a promising solution for real-world spectrum management and communication applications.

## 1. Introduction

Automatic modulation classification (AMC) plays a critical role in various fields, including electromagnetic spectrum management (Peng et al., 2021), radar systems (Li, 2020),

[1]Institute of Big Data Science and Industry, Key Laboratory of Evolutionary Science Intelligence of Shanxi Province, Shanxi University, Taiyuan, China [2]Shanxi Key Laboratory of Big Data Analysis and Parallel Computing, School of Computer Science and Technology, Taiyuan University of Science and Technology, Taiyuan, China. Correspondence to: Yuhua Qian <jinchengqyh@126.com>.

*Proceedings of the 42nd International Conference on Machine Learning*, Vancouver, Canada. PMLR 267, 2025. Copyright 2025 by the author(s).

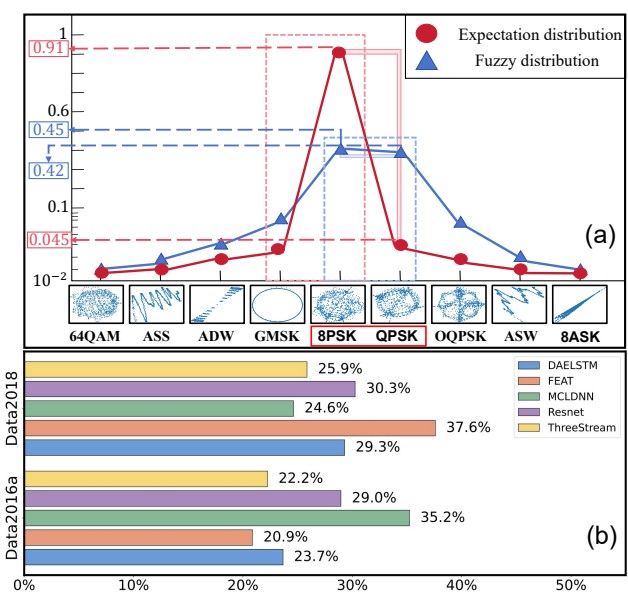

*Figure 1.* The blue line in (a) illustrates the prediction ambiguity, and the red line is the ideal case. (b) The sample proportion of prediction ambiguity in different models under different data sets is calculated, indicating that prediction ambiguity is not an accidental phenomenon.

unmanned aerial vehicle regulation (Guo et al., 2024), and biomedical signal processing (Gao et al., 2023). The goal of signal classification is to identify the target category of digital signals collected by physical sensing devices. Although data acquisition technology has advanced rapidly, the unique characteristics of signals make them highly susceptible to interference from various noise sources during the collection process. Consequently, effectively recognizing noisy data has become a prominent and widely discussed challenge today.

Deep learning has achieved remarkable success in various fields such as natural language processing (Zini & Awad, 2022), image processing (Qian & Fouhey, 2023) and multimodal learning (Liang et al., 2025). In recent years, it has also been introduced into signal classification tasks, significantly improving classification accuracy (Yuan et al., 2025; Zeng et al., 2024). However, with the continuous evolution of signal modulation methods, some modulation methods in the signal modulation category are only different

in the code bits. We plot the IQ of some modulation classes (see A.1) and find that there is little difference between these modulation modes with only different coding bits, such as 8PSK and QPSK. Models often exhibit low accuracy when recognizing such categories.

To investigate the classification challenges of these fine-grained modulation types, we analyze the Softmax values of the model's final layer, and observe an interesting phenomenon: the model's predictions for fine-grained samples are uncertain. For instance, in a nine-class classification task, when one model takes signal samples as input, and it outputs nine values representing the probabilities of the sample belonging to each of the nine categories. Among these probabilities, the difference between the top two rankings is very small. For example, the highest predicted probability is 0.45, while the second-highest is 0.42, as shown by the blue curve in Fig. 1(a). This phenomenon is referred to in this paper as the model *prediction ambiguity* phenomenon. We believe that the essential reason for this phenomenon is that the model has not learned effective features between similar classes. So, the model is easily confused when predicting such samples, and the prediction results are uncertain. Ideally, the distribution should resemble the red curve in Fig. 1(a), where category predictions are more definitive.

Unfortunately, *prediction ambiguity* phenomenon commonly exists in AMC. To illustrate the fact, we conducted classification tasks on the Data2016a and Data2018 datasets using five methods. These samples are characterized with IQ features with lengths of 128 and 1024, respectively. During the experiments, we counted the number of samples in the final training batch where the difference between the highest and second-highest probabilities was less than 0.3. Such samples were considered to exhibit prediction ambiguity. We also calculated the proportion of these samples relative to the entire training set. From the bar chart in Fig. 1(b), it is evident that approximately 25% of samples in each of the ten tasks exhibited prediction ambiguity, indicating that *prediction ambiguity phenomenon is not a coincidence*.

In this paper, we conducted a modeling analysis of this phenomenon, quantified the degree of prediction ambiguity in the model, and examined the relationship between prediction ambiguity and regularization gradients during training. When prediction ambiguity is pronounced, the regularization should return a larger gradient update to suppress the phenomenon. As prediction ambiguity diminishes, the regularization gradient should decrease correspondingly to ensure stable model training. Ultimately, we proposed a Fuzzy Regularization with an adaptive gradient mechanism to mitigate prediction ambiguity and guide the model toward learning more robust parameters in noisy environments.

In summary, ours main contributions are as follows:

1) We identify the prediction ambiguity phenomenon in automatic modulation classification and demonstrate that the model has the potential to achieve more optimal parameters by reducing ambiguity in its predictions.

2) We quantify the degree of prediction ambiguity and proposed Fuzzy Regularization (FR) with an adaptive gradient update mechanism. By penalizing model's prediction ambiguity, the model can be effectively guided toward learning more robust and optimal parameters.

3) We evaluate FR on the RADIOML 2016.10a, RADIOML 2016.10b, and RADIOML 2018.01A datasets. The results demonstrate that Fuzzy Regularization not only enhances the model's robustness but also improves its convergence speed to a certain extent.

## 2. Related Work

### 2.1. Automatic Modulation Classification

Automatic modulation recognition, as a key technology for managing the electromagnetic spectrum, has attracted increasing attention in recent years. Compared to previous methods based on hypothesis testing (Panagiotou et al., 2000; Wang & Wang, 2010; Xu et al., 2010) or feature engineering (Boutte & Santhanam, 2009; Su, 2013; Wu et al., 2008), deep learning-based automatic modulation methods show superior performance and can automatically construct effective discriminative features. Li et al. proposed a deep convolutional neural network based on AN-SF-CNN for very high frequency (VHF) radio signals (Li et al., 2018). Zhang et al. (Zhang et al., 2020) proposed a CNN-LSTM dual-stream architecture that effectively explores the interaction between time and spatial features. Xiao et al. (Xiao et al., 2023) designed a complex-valued network that directly models the complex relationships between complex-valued features. Zhang et al. (Zhang et al., 2023) introduced an automatic modulation classification method based on a multispectral attention mechanism from a frequency perspective. Peng et al. (Peng et al., 2018) transformed signal data into image representations for modulation classification. In summary, existing methods mainly focus on model architecture and model input. In contrast, we propose Fuzzy Regularization—a universally applicable regularization strategy that operates in the prediction space of automatic modulation recognition (AMR). This paradigm shift from architecture/input-centric to prediction-centric regularization establishes a new framework for building robust AMR systems in open-world scenarios.

### 2.2. Regularization

Regularization technique is frequently employed to better guide parameter optimization, ultimately reducing validation errors. Many strategies have been proposed typically

targeting domain-specific challenges. In the field of image recognition, to defend against adversarial attacks, Lee et al. (Lee et al., 2022) proposed a gradient diversity regularization to constrain and reduce gradient concentration, thereby building robust neural networks. Zhao et al. (Zhao et al., 2024) enhanced model performance by imposing penalties that encourage deterministic overall likelihood predictions. In multi-modal tasks, Ghahremani et al. (Ghahremani Boozandani & Wachinger, 2024) used the Frobenius norm to build a regularization-based batch normalization technique to reduce confounding effects and achieve independence between different modalities. For the open-set problem in automatic modulation recognition, Li et al. (Li et al., 2023) designed a margin prototype constraint to reduce the open space size by restricting the sample distribution range, thereby reducing the risk of open space. Additionally, Szegedy et al. (Szegedy et al., 2016) proposed a label smoothing regularization to alleviate the overfitting problem of the model. Compared with the aforementioned regularization methods, the fuzzy regularization proposed in this paper considers the impact of the regularization gradient during the model training process and design an adaptive gradient update mechanism.

## 3. The Proposed Method

This section primarily introduces the construction of the Fuzzy Regularization (FR) mechanism[1]. Subsection 3.1 presents a modeling analysis of the prediction ambiguity phenomenon, demonstrating that the model has the opportunity to learn better parameters by suppressing the ambiguity. Subsection 3.2 introduces how entropy and variance functions can be used to quantitatively measure the prediction ambiguity. Subsection 3.3 explains the implementation of the adaptive gradient mechanism and final definition of FR.

### 3.1. Problem Modeling

As mentioned in the introduction, when prediction ambiguity occurs in signal classification tasks, it indicates that the model has the potential to learn more robust parameters. This is because, for the cross-entropy loss function, when the model classifies correctly, the larger the maximum predicted probability for the correct class and the smaller the predicted probabilities for other classes, the lower the cross-entropy loss. So why does not the model further optimize this part of the loss?

We attempt to model and analyze this issue from a probabilistic statistics perspective. To simplify the modeling, we assume this is a binary classification problem with a single target. For a training batch $X = \{x_1, x_2, \ldots, x_k\}$

[1]The code is available at https://github.com/ruijiesang/FR-AMC.

with target label $Y = \{y_1, y_2\}$. Among them, $\frac{k}{2}$ samples are labeled as $y_1$ and $\frac{k}{2}$ labels are labeled as $y_2$. In addition during training the true labels are one-hot vectors and the model outputs are $[\hat{y}_1^{(i)}, \hat{y}_2^{(i)}]$, with $\hat{y}_j^{(i)}$ denoting the probability that the $i$th training sample belongs to class $j$. The corresponding cross-entropy loss ($\mathcal{L}_{batch}$) is:

$$\mathcal{L}_{batch} = \frac{1}{k} \sum_{i=1}^{k} \mathcal{L}^{(i)} = -\frac{1}{k} \sum_{i=1}^{k} \sum_{j=1}^{2} y_j^{(i)} \log(\hat{y}_j^{(i)})$$

$$= -\frac{1}{k} \left( \sum_{i=1}^{\frac{k}{2}} \log(\hat{y}_1^{(i)}) + \sum_{i=\frac{k}{2}+1}^{k} \log(\hat{y}_2^{(i)}) \right)$$

$$= -\frac{1}{k} \left( \sum_{i=1}^{\frac{k}{2}} \log(\hat{y}_1^{(i)}) + \sum_{i=\frac{k}{2}+1}^{k} \log(1 - \hat{y}_1^{(i)}) \right). \quad (1)$$

Define $p_i$ is an independent and identically distributed random variable, represents the highest probability value predicted by the model for each category of the $i$th sample ($p_i = Max(\hat{y}_1^{(i)}, \hat{y}_2^{(i)})$). For a batch of $k$ samples in which $m$ samples are misclassified, the corresponding cross-entropy loss ($\mathcal{L}_{batch}^{(m)}$) can be expressed as:

$$\mathcal{L}_{batch}^{(m)}(p_i) = -\frac{1}{k} \left( \sum_{i=1}^{m} \log(1 - p_i) + \sum_{i=m+1}^{k} \log(p_i) \right). \quad (2)$$

If the probability of the model classifying a single sample correctly is $\alpha$, then the probability of classification error is $1 - \alpha$. In a binary classification task involving $k$ samples, the probability that the model misclassifies $m$ samples is:

$$\mathrm{P}_{batch}^{(m)}(\alpha) = C_k^m (1 - \alpha)^m (\alpha)^{k-m}. \quad (3)$$

Combining Eqs. (2) and (3), we construct the following expected loss model:

$$\mathrm{E}(\mathcal{L}_{batch}) = \sum_{m=0}^{k} \mathrm{P}_{batch}^{(m)}(\alpha) \mathcal{L}_{batch}^{(m)}(p_i). \quad (4)$$

Due to the model's poor discriminative ability between these two classes, the classification accuracy $\alpha$ can be assumed to fluctuate around 0.5. According to the law of large numbers, as the number of trials increases, the sample mean of the random variable will converge to its theoretical expectation, thus $\alpha = 0.5$. Moreover, during the same training batch, the model's performance is unlikely to change significantly. This implies that the highest value of the predicted probability tend to stabilize and can be approximated by a constant $u$ ($0 < u < 1$). Consequently, it can further be assumed that $p_i = u$. Based on these assumptions, Eq. (4) can be simplified as follows:

$$\mathrm{E}(\mathcal{L}_{batch}) = \sum_{m=0}^{k} \mathrm{P}_{batch}^{(m)}\left(\frac{1}{2}\right) \mathcal{L}_{batch}^{(m)}(u), \quad (5)$$

where $\mathcal{L}_{batch}^{(m)}(u) = -\frac{1}{k}(m\log(1-u) + (k-m)\log(u))$,
$P_{batch}^{(m)}(\frac{1}{2}) = (\frac{1}{2})^k C_k^m$.

By reorganizing and combining Eq. (5), Eq. (4) can be further expanded as:

$$
\begin{aligned}
E(\mathcal{L}_{batch}) &= \sum_{m=0}^{k} P_{batch}^{(m)}\left(\frac{1}{2}\right) \mathcal{L}_{batch}^{(m)}(u) \\
&= Q^*(\log(u) + log(1-u))) \\
&= Q^*(\log(u - u^2))),
\end{aligned}
\tag{6}
$$

where $Q^* = -\frac{1}{k}(\frac{1}{2})^k (\sum_{i=0}^{k} C_k^i (k-i))$.

During the training process, the model adjusts its parameters by optimizing the objective function $\arg\min_u E(\mathcal{L}_{batch})$. We observe that when $u = 0.5$, the expected loss $E(\mathcal{L}_{batch})$ reaches its minimum. We believe this result occurs because the model's discriminative ability is weak when dealing with similar classes. Even after multiple iterations, the model fails to effectively extract the critical discriminative features of these similar classes. As a result, the model exhibits a "lazy" behavior, tending to smooth the predicted probability distribution, which leads to prediction fuzziness. As seen in the above formula, by fuzzifying the prediction distribution, the model can effectively control the growth of the loss. To alleviate this issue, we propose Fuzzy Regularization (FR). When the model exhibits weak discriminative power for similar classes and shows prediction fuzziness, FR introduces an additional penalty term. This term forces the model to learn the relationships between the top k predicted classes, thereby encouraging the model to learn better parameters and improve its robustness.

### 3.2. Quantifying Ambiguity Degree of Predict Results

To optimize the prediction fuzziness issue, we first need to quantify and define the degree of fuzziness in the model's predictions. The degree of fuzziness essentially reflects the level of disorder in the model's predictive distribution, which represents the degree of informational disorder. Therefore, one can naturally think of entropy functions and L2 norms, both of which can describe the degree of disorder in the information. Thus, we can use entropy functions and L2 norms to quantify the degree of prediction fuzziness.

Specifically, for the entropy function, when the prediction is fuzzy, meaning the information is more disordered, it returns a larger value. When the prediction distribution is more certain, meaning the information is more definite, the entropy function returns a smaller value. For the L2 norm, when the prediction is fuzzy, meaning the predicted probability distribution is more spread out, the L2 norm returns a smaller value. When the prediction fuzziness is not significant, meaning the predicted probability values are more concentrated, the L2 norm returns a larger value.

Let $M$ denote the prediction fuzzy loss for a $c$-class classification problem, where higher values of $M$ indicate greater ambiguity in the model's predictions. We can quantify it using either the entropy function or the L2 norm. The two formulations are given as follows:

$$
M = -\sum_{j=1}^{c} \hat{y}_j \log(\hat{y}_j),
\tag{7}
$$

$$
M = -\sum_{j=1}^{c} (\hat{y}_j - \frac{1}{c})^2,
\tag{8}
$$

where $\hat{y}_j$ represents the model's prediction probability value that the sample may belong to each class.

### 3.3. Adaptive Adjustment of the Update Gradient

After constructing the quantitative representation of the prediction ambiguity, we attempt to directly add Eq. (7) or Eq. (8) from Section 3.2 as a regularization term to the loss function, aiming to impose an additional penalty when prediction ambiguity occurs, in order to suppress this phenomenon. However, the experimental results were not as expected. Upon analysis, we found that this issue might be due to the gradient update strength of the entropy function or L2 norm. See A.2 for detailed analysis. The final definition of fuzzy regularization (FR) is as follows:

- When the prediction is ambiguous, the loss function should return a larger value; when the prediction ambiguity is not significant, the loss function should return a smaller value.

- When the prediction is ambiguous, the gradient of the loss function should correspond to a larger value; when the prediction ambiguity is not significant, the gradient of loss function should correspond to a smaller value.

Under the binary classification problem, we assume that the probability value of the first category is $p$, then the probability value of the second category is $1 - p$, and the coordinate system is established with the prediction probability value of the first category $p$ as the horizontal coordinate and the degree of prediction ambiguity as the vertical coordinate. The red dot indicates the most serious prediction ambiguity.

That is, we expect the gradient of the regularization term to have a shape similar to $y = -\frac{1}{x}$, as shown in Fig. 2(a). By integrating $y = -\frac{1}{x}$, we obtain $y = -\log(|x| - 0.5)$. To reduce the computational complexity, we use the L2 norm as the quantitative measure of model prediction ambiguity. Meanwhile, to eliminate the influence of the model's predicted distribution on $M$ (for example, when the two largest prediction probabilities are [0.3, 0.3] or [0.4, 0.4], both indicating significant prediction ambiguity, but the resulting $M$ values are different), we focus solely on the distribution state

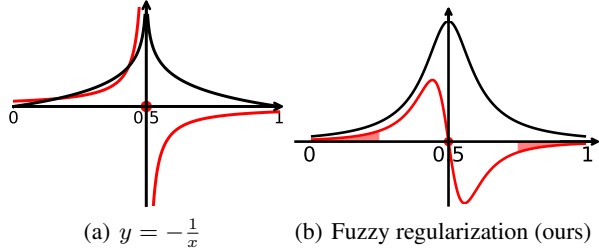

(a) $y = -\frac{1}{x}$     (b) Fuzzy regularization (ours)

*Figure 2.* Ideal curve and its derivative are shown in black and red, respectively, in (a); FR curve and its derivative in (b). $x$ and $y$ axises denote the category probability value and fuzziness degree.

and normalize $M$. Based on this, the prediction ambiguity loss for a single sample is given as follows:

$$\text{Loss} = \log\left(\frac{\sum_{j=1}^{k} \hat{y}_j^2 - k \times \left(\frac{\sum_{j=1}^{k} \hat{y}_j}{k}\right)^2}{\left(\sum_{j=1}^{k} \hat{y}_j\right)^2 - k \times \left(\frac{\sum_{j=1}^{k} \hat{y}_j}{k}\right)^2}\right). \quad (9)$$

Here, $\hat{y}_j$ represents the model's predicted probability for each class, $\sum_{j=1}^{k} \hat{y}_j^2$ represents the L2 norm of the sample, $k \times \left(\frac{\sum_{j=1}^{k} \hat{y}_j}{k}\right)^2$ represents the minimum L2 norm when considering the top $k$ values. For instance, when $k=3$, if the predicted distribution is $\left(\frac{\sum_{j=1}^{3} \hat{y}_j}{3}, \frac{\sum_{j=1}^{3} \hat{y}_j}{3}, \frac{\sum_{j=1}^{3} \hat{y}_j}{3}\right)$, the prediction ambiguity is most severe. $\left(\sum_{j=1}^{k} \hat{y}_j\right)^2$ represents the maximum L2 norm when considering the top $k$ values. For example, when $k=3$, if the predicted distribution is $(\sum_{j=1}^{k} \hat{y}_j, 0, 0)$, there is no prediction ambiguity. $\log(*)$ ensures the adaptive adjustment of the regularization gradient.

In the training process, batch training is typically used, where each batch consists of multiple samples, and a single loss value is returned for the batch. The loss function described above only reflects the prediction distribution of an individual sample, without fully utilizing the information between samples to guide the model. To address this, we integrate both individual sample information and the overall information of the entire batch using a log-normal distribution function. Our final FR is defined as follows:

$$\text{F}(\hat{y}) = \frac{\sigma}{\text{T}\sqrt{2\pi}} \exp\left[-\frac{\sigma^2}{2} \log(\text{T})^2\right], \quad (10)$$

$$s.t. \begin{cases} \sigma = -\frac{1}{N} \sum_{i=1}^{N} \frac{\sum_{j=1}^{k} \hat{y}_j^{(i)} \log(\hat{y}_j^{(i)})}{\log(\tau)}, \\ \text{T} = \frac{\sum_{j=1}^{k} \left(\hat{y}_j^{(i)}\right)^2 - k\left(\frac{\sum_{j=1}^{k} \hat{y}_j^{(i)}}{k}\right)^2}{\left(\sum_{j=1}^{k} \hat{y}_j^{(i)}\right)^2 - k\left(\frac{\sum_{j=1}^{k} \hat{y}_j^{(i)}}{k}\right)^2}. \end{cases} \quad (11)$$

where $\hat{y}$ denotes the output of the model, $\hat{y}_j^{(i)}$ denotes the probability that the $i$th sample belongs to class $j$, $N$ denotes the sample size of the batch, $\tau$ denotes the $\tau$ classification task and $k$ denotes the selection of the first $k$ values.

*Table 1.* Performance comparison with best-performing methods.

| Datasets | Methods | F1-Score | ACC | H_ACC |
|---|---|---|---|---|
| **Data2016a** | DAELSTM | 0.7766 | 78.55% | 80.84% |
| | FEAT | 0.7544 | 76.67% | 78.31% |
| | MCLDNN | 0.6772 | 69.66% | 70.98% |
| | ThreeStream | 0.7683 | 78.11% | 80.35% |
| | Resnet | 0.7347 | 74.48% | 77.29% |
| | **Ours** | **0.8091** | **81.68%** | **84.90%** |
| **Data2016b** | DAELSTM | 0.9139 | 91.63% | 93.02% |
| | FEAT | 0.8186 | 83.10% | 83.66% |
| | MCLDNN | 0.9065 | 91.16% | 92.02% |
| | ThreeStream | 0.9035 | 90.76% | 92.34% |
| | Resnet | 0.8919 | 89.52% | 90.69% |
| | **Ours** | **0.9210** | **92.62%** | **93.58%** |
| **Data2018** | DAELSTM | 0.7535 | 75.81% | 89.33% |
| | FEAT | 0.7129 | 72.33% | 79.52% |
| | MCLDNN | 0.7895 | 80.28% | 91.62% |
| | ThreeStream | 0.8197 | 82.08% | 93.74% |
| | Resnet | 0.7267 | 73.67% | 86.46% |
| | **Ours** | **0.8666** | **86.75%** | **96.62%** |

As illustrated in Fig. 2(b), the black curve represents the Fuzzy Regularization (FR) function, while its derivative is shown in red. The derivative of FR does not fully satisfy the second part of the definition, but it does not contradict our intention. Because the prediction ambiguity will gradually weaken under the supervision of FR, we only hope that the gradient of FR will decrease with the weakening of the fuzziness in the case that the prediction ambiguity is not very serious (the red shaded part).

## 4. Experiments

In this section, we show the advantages of the Fuzzy Regularization (FR) from effectiveness, generalizability, robustness, training behaviour, and parameter sensitivity.

### 4.1. Experiments Settings

Our datasets are primarily derived from publicly available wireless modulation type recognition datasets. We evaluated the effectiveness of Fuzzy Regularization (FR) in suppressing prediction ambiguity using five models[2]: ResNet (O'Shea et al., 2018), DAELSTM (Ke & Vikalo, 2021), MCLDNN (Xu et al., 2020), Three-Stream (Liang et al., 2021), and FEAT (Chen et al., 2023) on the RadioML 2016.10a (Data2016a) (O'shea & West, 2016), RadioML 2016.10b (Data2016b) (A.3), and RadioML 2018.01A (Data2018) (O'Shea et al., 2018) signal datasets. The evaluation was based on three metrics: F1-Score, ACC, and H-ACC. Detailed descriptions of datasets (A.3), the methods (A.4) and evaluation metrics (A.5) can be found in the appendix.

---

[2]The code can be found https://github.com/DTMB-DL/TransGroupNet.

*Table 2.* Comparison of methods across datasets.

| Methods | Data2018 | | | Data2016a | | | Data2016b | | |
|---|---|---|---|---|---|---|---|---|---|
| | F1-Score | ACC | H_ACC | F1-Score | ACC | H_ACC | F1-Score | ACC | H_ACC |
| DAELSTM | 0.7535 | 75.81% | 89.33% | 0.7766 | 78.55% | 80.84% | 0.9139 | 91.63% | 93.02% |
| DAELSTM+FR | **0.7684** | **78.08%** | **92.02%** | **0.8091** | **81.68%** | **84.90%** | **0.9157** | **92.03%** | **93.22%** |
| Δ% | +1.49% | +2.27% | +2.69% | +3.25% | +3.13% | +4.06% | +0.18% | +0.40% | +0.20% |
| FEAT | 0.7129 | 72.33% | 79.52% | 0.7544 | 76.67% | 78.31% | 0.8186 | 83.10% | 83.66% |
| FEAT+FR | **0.7275** | **74.36%** | **81.89%** | **0.7713** | **78.25%** | **79.62%** | **0.8295** | **83.69%** | **84.45%** |
| Δ% | +1.46% | +2.03% | +2.37% | +1.69% | +1.58% | +1.31% | +1.09% | +0.59% | +0.79% |
| MCLDNN | 0.7895 | 80.28% | 91.62% | 0.6772 | 69.66% | 70.98% | 0.9065 | 91.16% | 92.02% |
| MCLDNN+FR | **0.8026** | **81.48%** | **92.29%** | **0.7237** | **73.79%** | **75.32%** | **0.9210** | **92.62%** | **93.58%** |
| Δ% | +1.31% | +1.20% | +0.67% | +4.65% | +4.13% | +4.34% | +1.45% | +1.46% | +1.56% |
| ThreeStream | 0.8197 | 82.08% | 93.74% | 0.7683 | 78.11% | 80.35% | 0.9035 | 90.76% | 92.34% |
| ThreeStream+FR | **0.8666** | **86.75%** | **96.62%** | **0.7938** | **80.37%** | **83.04%** | **0.9119** | **91.67%** | **92.65%** |
| Δ% | +4.69% | +4.67% | +2.88% | +2.55% | +2.26% | +2.69% | +0.84% | +0.91% | +0.31% |
| Resnet | 0.7267 | 73.67% | 86.46% | 0.7347 | 74.48% | 77.29% | 0.8919 | 89.52% | 90.69% |
| Resnet+FR | **0.7442** | **74.98%** | **87.39%** | **0.7500** | **75.81%** | **77.99%** | **0.8997** | **90.43%** | **91.44%** |
| Δ% | +1.75% | +1.31% | +0.93% | +1.53% | +1.33% | +0.70% | +0.78% | +0.91% | +0.75% |

## 4.2. Comparison with Other Methods

In this section, we respectively selected the best-performing models on Data2016a, Data2016b and Data2018 datasets, namely DAELSTM, DAELSTM and ThreeStream. Verify whether it is possible to further enhance the performance of the best-performing models by adding FR. As shown in Table 1, our method outperforms the current best-performing models across all three datasets. On the Data2016a, Data2016b, and Data2018 datasets, our method shows a 1%-4% performance improvement over the best-performing models in both ACC and H-ACC.

It is also found that the performance improvement brought by the FR regularization varies across different datasets. This phenomenon might be attributed to the fact that the existing SOTA methods already achieve high performance on the Data2016b dataset, with a peak average accuracy of 91.63%. In contrast, the highest average accuracy on the Data2016a and Data2018 datasets is only about 80%. This indicates that the SOTA models on the Data2016b dataset have already identified a relatively optimal classification plane, resulting in a smaller corrective effect when the FR regularization is applied.

Across all three datasets, our method outperforms the compared models in both average accuracy and peak accuracy, demonstrating that the FR term can effectively refine the classification plane, thereby improving model performance.

## 4.3. Generalizability of the Fuzzy Regularization (FR)

This section aims to validate the generalizability of the proposed FR by integrating it into five state-of-the-art (SOTA) methods. To ensure the fairness and reliability of the experiments, all controllable parameters—including the learning rate, random seeds, and model initialization—were kept

consistent before and after applying FR.

As demonstrated in Table 2, the methods enhanced with FR consistently outperform their counterparts across all datasets and performance metrics. Specifically, for the same task (i.e., the same model and dataset), the maximum improvement in accuracy (ACC) is observed from 69.66% to 73.79%, while the harmonic accuracy (H-ACC) increases from 70.98% to 75.32%, both achieving an approximate 4% enhancement. Conversely, the smallest improvement is noted for ACC, which rises from 91.63% to 92.03%, and H-ACC, which increases from 93.02% to 93.22%, with an increment of only 0.3%.

To further analyze the variability in performance improvements across tasks, we investigated the correlation between the effectiveness of FR and the proportion of samples exhibiting prediction ambiguity in tasks without FR. Notably, for the task with the highest performance improvement, 35.2% of the samples exhibited prediction ambiguity, whereas for the task with the lowest improvement, this proportion was only 5.14%. This finding suggests that FR provides greater performance gains for tasks with more severe prediction ambiguity, highlighting its ability to address such challenges effectively.

In summary, the experimental results comprehensively validate the general applicability and effectiveness of FR across diverse datasets, particularly in scenarios where prediction ambiguity is prevalent.

## 4.4. Robustness of the Fuzzy Regularization (FR)

Post-deployment automatic modulation recognition models inevitably encounter interference from the different environment from the training data. Therefore, enhancing the model's robustness becomes a critical consideration. Theo-

*Table 3.* Performance comparison on data sets with different noise factors.

| Datasets | Methods | Noisy Factor_20% | | | Noisy Factor_40% | | | Noisy Factor_60% | | |
|---|---|---|---|---|---|---|---|---|---|---|
| | | F1-Score | ACC | H_ACC | F1-Score | ACC | H_ACC | F1-Score | ACC | H_ACC |
| Noise2016a | DAELSTM | 0.7662 | 77.18% | 79.66% | 0.6918 | 71.41% | 74.17% | 0.5733 | 60.04% | 63.50% |
| | **DAELSTM+FR** | **0.7754** | **79.61%** | **82.82%** | **0.7220** | **73.14%** | **75.64%** | **0.6322** | **63.51%** | **66.45%** |
| | FEAT | 0.7558 | 76.66% | 78.08% | 0.5648 | 58.80% | 59.90% | 0.5398 | 56.38% | 58.08% |
| | **FEAT+FR** | **0.7868** | **79.80%** | **80.87%** | **0.5675** | **59.27%** | **59.79%** | **0.5487** | **56.79%** | **58.08%** |
| | MCLDNN | 0.6615 | 68.62% | 70.09% | 0.5518 | 57.85% | 59.28% | 0.5294 | 56.03% | 58.20% |
| | **MCLDNN+FR** | **0.7027** | **71.44%** | **72.98%** | **0.5679** | **58.90%** | **60.41%** | **0.5408** | **56.67%** | **59.13%** |
| | ThreeStream | 0.6749 | 70.76% | 73.13% | 0.6436 | 65.63% | 69.08% | 0.5315 | 53.13% | 58.25% |
| | **ThreeStream+FR** | **0.7789** | **78.90%** | **81.36%** | **0.7140** | **72.12%** | **75.76%** | **0.6017** | **61.06%** | **64.80%** |
| | Resnet | 0.7282 | 73.75% | 76.28% | 0.6838 | 68.94% | 72.01% | 0.5971 | 60.21% | 63.28% |
| | **Resnet+FR** | **0.7393** | **74.70%** | **76.59%** | **0.6992** | **70.68%** | **72.60%** | **0.6326** | **64.24%** | **66.96%** |
| Noise2016b | DAELSTM | 0.8899 | 89.36% | 90.95% | 0.8147 | 81.81% | 83.73% | 0.6660 | 68.55% | 70.92% |
| | **DAELSTM+FR** | **0.8932** | **89.86%** | **91.13%** | **0.8196** | **82.75%** | **84.56%** | **0.6890** | **69.79%** | **72.29%** |
| | FEAT | 0.8177 | 82.65% | 83.68% | 0.7656 | 78.81% | 79.79% | 0.6839 | 71.72% | 72.99% |
| | **FEAT+FR** | **0.8166** | **82.84%** | **83.75%** | **0.7941** | **80.36%** | **81.52%** | **0.7246** | **73.97%** | **75.49%** |
| | MCLDNN | 0.8406 | 85.26% | 86.47% | 0.7164 | 76.21% | 77.52% | 0.5410 | 59.02% | 60.95% |
| | **MCLDNN+FR** | **0.9028** | **90.73%** | **91.71%** | **0.8115** | **81.91%** | **83.30%** | **0.7245** | **74.31%** | **75.91%** |
| | ThreeStream | 0.8500 | 85.91% | 87.25% | 0.7149 | 75.86% | 78.06% | 0.5270 | 57.13% | 60.14% |
| | **ThreeStream+FR** | **0.8719** | **87.66%** | **88.78%** | **0.7605** | **78.70%** | **80.57%** | **0.6834** | **69.95%** | **72.20%** |
| | Resnet | 0.8654 | 87.02% | 88.17% | 0.7887 | 80.15% | 81.29% | 0.6790 | 70.02% | 71.98% |
| | **Resnet+FR** | **0.8665** | **87.24%** | **88.33%** | **0.8036** | **81.09%** | **82.57%** | **0.7038** | **72.09%** | **73.91%** |
| Noise2018 | DAELSTM | 0.6846 | 70.11% | 81.47% | 0.5563 | 58.04% | 66.50% | 0.4407 | 46.98% | 54.24% |
| | **DAELSTM+FR** | **0.7049** | **71.03%** | **81.75%** | **0.5826** | **59.40%** | **68.22%** | **0.4673** | **48.27%** | **55.71%** |
| | FEAT | 0.6205 | 64.20% | 70.48% | 0.5341 | 56.53% | 62.20% | 0.4766 | 50.18% | 55.85% |
| | **FEAT+FR** | **0.6369** | **65.99%** | **73.30%** | **0.5742** | **59.19%** | **65.32%** | **0.4847** | **51.03%** | **56.40%** |
| | MCLDNN | 0.6826 | 70.12% | 78.82% | 0.4829 | 50.02% | 56.14% | 0.2193 | 27.25% | 30.23% |
| | **MCLDNN+FR** | **0.7241** | **73.33%** | **82.71%** | **0.5043** | **53.43%** | **60.82%** | **0.2417** | **29.93%** | **33.06%** |
| | ThreeStream | 0.7387 | 74.40% | 84.12% | 0.4475 | 46.28% | 49.82% | 0.0804 | 12.93% | 14.46% |
| | **ThreeStream+FR** | **0.7600** | **76.38%** | **85.58%** | **0.4429** | **46.52%** | **50.24%** | **0.1613** | **19.90%** | **22.74%** |
| | Resnet | 0.6330 | 64.75% | 74.06% | 0.5199 | 54.06% | 60.49% | 0.4222 | 44.16% | 49.43% |
| | **Resnet+FR** | **0.6370** | **64.86%** | **74.30%** | **0.5359** | **54.89%** | **61.25%** | **0.4314** | **45.75%** | **51.42%** |

retically, FR improves robustness by suppressing ambiguity through reducing intra-class distances and enlarging inter-class distances, thereby enhancing the model's performance.

To empirically validate whether FR enhances model robustness, we conducted a series of experiments. When data are perturbed, smaller intra-class distances and larger inter-class distances ensure that the projections of perturbed samples on the classification plane remain within a controllable range, thereby reducing the likelihood of misclassification. To this end, we designed experiments using the Data2016a, Data2016b, and Data2018 datasets, generating corresponding noisy datasets named Noise2016a, Noise2016b, and Noise2018. The detailed noise generation process is described in Appendix A.4.

As shown in Table 3, the robustness of models equipped with FR modules is significantly improved across different tasks. For instance, MCLDNN-FR achieves a performance improvement of up to 15% (from 59.02% to 74.31%) on Noise2016b under a 60% noisy factor, increasing accuracy. For other tasks, performance gains of 1%-3% are consistently observed. These results demonstrate that FR effec-

tively identifies classification planes with smaller intra-class distances and larger inter-class distances, thereby enhancing robustness against noisy data.

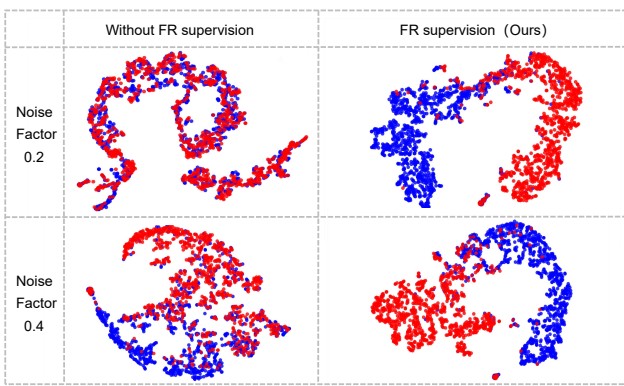

*Figure 3.* T-SNE intuitively shows that two similar modulation classes can be effectively distinguished by adding FR regularization. (Red dots represent 8PSK and blue dots represent QPSK)

Furthermore, to visually illustrate the impact of FR on the

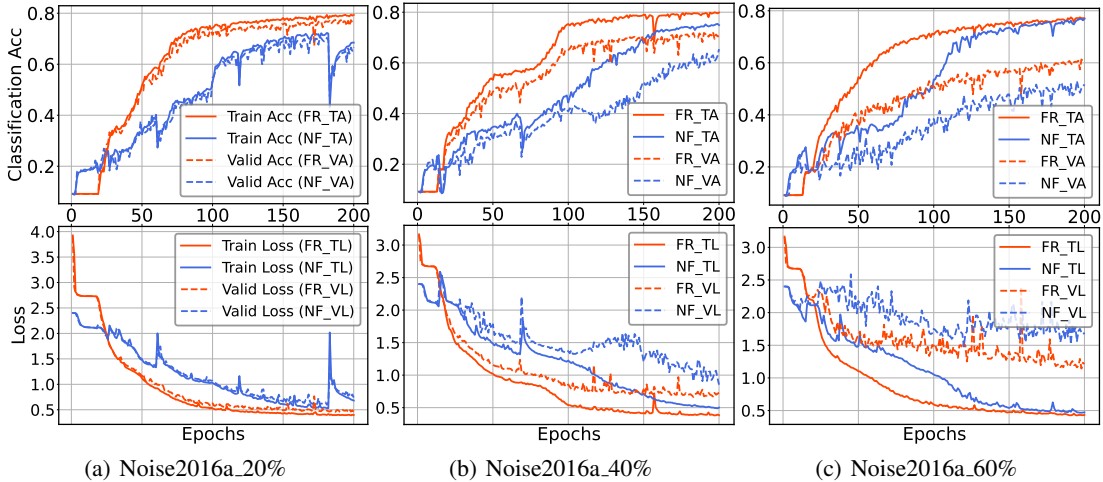

*Figure 4.* It presents the accuracy curves and loss curves on different noise datasets when there is FR and when there is NO FR(NF).

model's classification hyperplane, we selected two similar modulation classes, QPSK and 8PSK, as discussed in Section 1. The same baseline model was trained with and without FR regularization, and the final layer outputs of the two classes were visualized using t-SNE. As shown in Fig. 3, the classification hyperplane under FR supervision exhibits clearer separation between classes compared to the one without FR regularization, confirming its effectiveness in encouraging margin maximization between confusable modulation clusters.

In summary, both quantitative metrics and visualization results validate that FR significantly enhances the robustness of automatic modulation recognition models for noisy data.

### 4.5. Training Behaviour Analysis

This section elucidates the mechanisms behind the accelerated convergence and stabilized training trajectories induced by Fuzzy Regularization (FR).

As shown in Fig. 4, the FR-regularized model (red curve) achieves both faster convergence and smoother training trajectories compared to the baseline (blue curve). We attribute this dual improvement to FR's ambiguity-aware curriculum learning mechanism: (1) *Early-Stage Prediction Sharpening.* FR induces rapid entropy reduction for high-confidence predictions during initial training phases. Sharpening the probability distribution of correctly classified samples leads to their prediction distribution becoming more certain early, reducing further optimization for those samples. This aligns with curriculum learning principles (Bengio et al., 2009), where easier patterns are mastered before complex ones. (2) *Dynamic Hard Sample Emphasis.* Sharpening the prediction distribution for incorrect samples makes the errors more pronounced, causing the cross-entropy function to fo-

cus more on those incorrectly predicted samples, allowing for quicker correction. Thus, under FR supervision, the model converges faster. It can also be observed that the red curve represents smoother overall training.

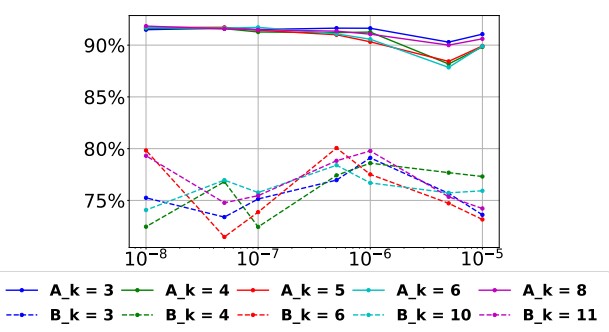

*Figure 5.* The horizontal coordinate is $\gamma$ , which represents the scaling factor before FR, A_k represents the selection of parameter $k$ under the Data2016a dataset, and B_k represents the selection of parameter $k$ under the Data2016b dataset.

### 4.6. Parameter Sensitivity Analysis

In this section, we will discuss how to quickly determine the satisfying values for two hyperparameters when using FR across different tasks. Fig. 5 illustrates the model accuracy achieved by the ThreeStream+FR with various parameter combinations on the Data2016a and Data2016b datasets. Through comprehensive analysis, it is can be seen that the optimal value of $\gamma$ is related to the magnitude of the initial FR compared to the cross-entropy loss. Specifically, when the initial FR is approximately two orders of magnitude smaller than the cross-entropy loss, the model tends to achieve peak performance. Regarding the parameter $k$, our analysis reveals that the parameter $k$ is intrinsically linked to the number of semantically similar classes associated

with each category in the dataset. For instance, if a class has $m$ semantically related classes, the output activations corresponding to these $m + 1$ classes tend to be the most proximate. Consequently, setting $k$ to the cardinality of the semantically similar class set generally yields optimal performance. These insights provide valuable guidance for parameter tuning in FR applications.

## 5. Conclusion

In this paper, we start from the prediction ambiguity phenomenon that arises in deep learning-based automatic modulation recognition tasks, providing a theoretical derivation for this phenomenon. We then design the FR method to suppress the occurrence of prediction ambiguity, thereby guiding the model to find a better and more robust decision boundary. When designing the FR regularization, we also consider the impact of the gradient of the regularization term on model training, so we introduce an adaptive gradient mechanism. As the prediction ambiguity gradually diminishes, the absolute value of the update gradient of the FR decreases accordingly, ensuring that the model gradually converges during training. Furthermore, the experimental section demonstrates the effectiveness and generality of our method, and verifies that the model under FR supervision exhibits improved robustness when confronted with noisy data. In the future, we will focus on exploring the more effective fuzzy regularization forms.

## Acknowledgements

This work was supported by National Natural Science Foundation of China (Nos. T2495251, 62406218, 62306171, T2495253, 62476160, 62376146), the Science and Technology Major Project of Shanxi (No. 202201020101006), and Fundamental Research Program of Shanxi Province (No. 202203021222183).

## Impact Statement

This paper presents work whose goal is to advance the field of Machine Learning. There are many potential societal consequences of our work, none which we feel must be specifically highlighted here.

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

# A. Appendix

In the supplemental material:

- **A.1.** Signal Visualization.

- **A.2.** Regular Gradient Problem.

- **A.3.** Datasets.

- **A.4.** Compared Methods.

- **A.5.** Evaluation Metrics.

## A.1. Signal Visualization

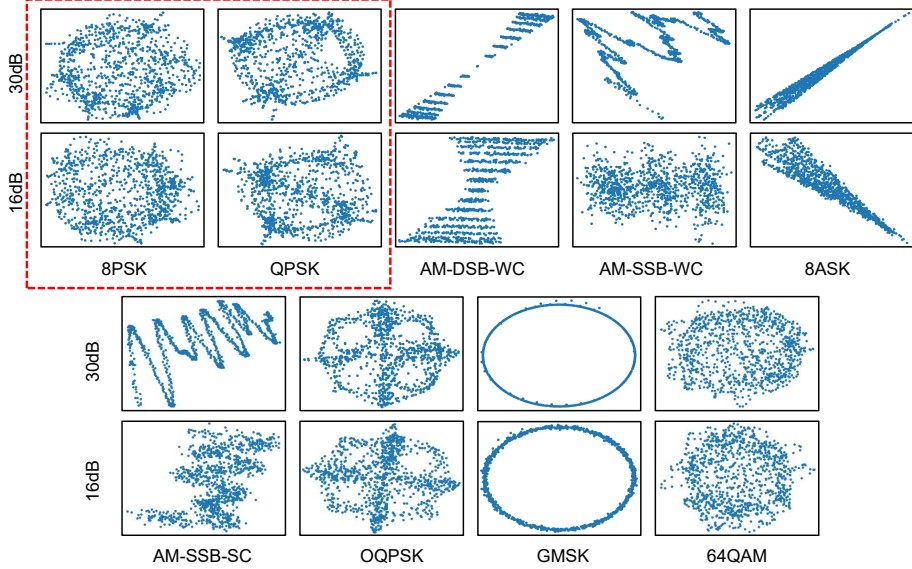

*Figure 6.* An IQ map of partial automatic modulation recognition under 16dB and 30dB, with 8PSK and QPSK in the red dashed line representing two more similar modulation classes.

## A.2. Regular Gradient Problem

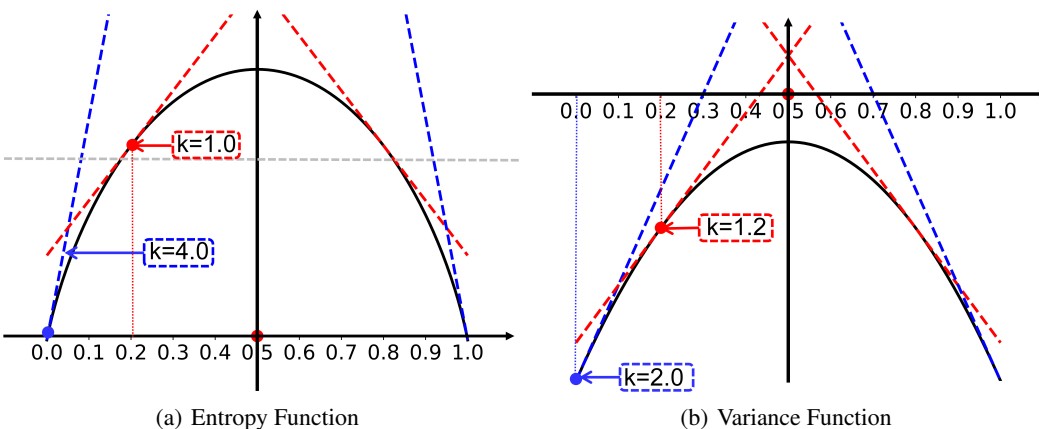

(a) Entropy Function                    (b) Variance Function

*Figure 7.* The horizontal coordinate represents the prediction probability value that the model considers the sample to belong to the first category under the binary classification task, and the vertical coordinate represents the degree of prediction ambiguity. (a) means that entropy function is used to measure the degree of prediction ambiguity; (b) means that variance function is used to measure the degree of prediction ambiguity.

As shown in the Fig.7, when the prediction ambiguity is not obvious, that is, when the model is more confident in the prediction results, the update gradient of the entropy function or variance function still corresponds to a large value, which leads to a large degree of change in the parameters of the model when the distribution of the predicted probability value of the model reaches our ideal distribution, resulting in unstable model training and difficulty in finding the optimal solution.

### A.3. Datasets

Our datasets mainly come from publicly available wireless modulation recognition datasets. We evaluated the effectiveness of FR in suppressing prediction ambiguity on six signal datasets: RadioML2016.10a, RadioML2016.10b, RADIOML 2018.01 A and their corresponding noise versions including Noise2016a, Noise2016b and Noise2018.

- RADIOML 2016.10a: This dataset contains 220,000 signal samples, consisting of 11 types of wireless modulation (eight digital modulations and three analog modulations). The eight digital modulations are BPSK, QPSK, 8PSK, 16QAM, 64QAM, BFSK, CPFSK, and PAM4. The three analog modulations are WB-FM, AM-SSB, and AM-DSB. The signal-to-noise ratio (SNR) ranges from -20dB to 18dB, with 20 different SNR levels. For each SNR and modulation type, there are 1,000 signal samples, each with a sample length of 128.

- RADIOML 2016.10b: This dataset contains 1,200,000 signal samples, consisting of 10 wireless modulations. The modulation classes are 8PSK, AM-DSB, BPSK, CPFSK, GFSK, PAM4, QAM16, QAM64, QPSK and WBFM. The signal-to-noise ratio ranges from -20db to 18db, with 20 different signal-to-noise ratios. There are 6000 signal samples with the same SNR and the same modulation category respectively, and the sampling length of each signal sample is 128.The dataset can be accessed for download from the following URL: https://www.deepsig.io/datasets.

- RADIOML 2018.01 A (O'Shea et al., 2018): This dataset consists of 2,555,904 signal samples, encompassing 24 types of wireless modulation. The modulation types include OOK, 4ASK, 8ASK, BPSK, QPSK, 8PSK, 16PSK, 32APSK, 64APSK, 128APSK, 16QAM, 32QAM, 64QAM, 128QAM, 256QAM, AM-SSB-WC, AM-SSB-SC, AM-DSB-WC, AM-DSB-SC, FM, GMSK, and OQPSK. The signal-to-noise ratio (SNR) ranges from -20dB to 30dB, with 26 different SNR levels. For each SNR and modulation type, there are 4,096 signal samples, each with a sample length of 1,024.

- Noise2016a, Noise2016b, Noise2018: To validate the impact of FR on model robustness under noisy data conditions, we generated signal data with different noise intensities based on the three datasets. Specifically, we first designed an adjustable scaling factor to control the noise intensity. Then, we calculated the standard deviation of each sample in the datasets and multiplied it by the scaling factor to determine the intensity of the white noise for each sample. Subsequently, white noise was generated for each sample based on the calculated intensity. Finally, the generated white noise was added to the original sample to obtain the final noisy data.

It is noted that adding noise to low-SNR data would significantly degrade data quality, making the model's recognition performance on such low-SNR noisy data extremely poor and thus lacking analytical significance. Therefore, in our experiments, we only used data with an SNR of 0 or higher from the aforementioned three datasets.

### A.4. Compared Methods

For the baseline models in our experiments, we selected five state-of-the-art models commonly used in deep learning and signal classification tasks to evaluate the effectiveness of FR regularization. These models include ResNet, DAELSTM, MCLDNN, Three-Stream, and FEAT. These five models encompass several critical neural network architectures, such as convolutional neural networks, recurrent neural networks, multi-stream networks, and Transformer-based networks, ensuring diversity across the selected models. Below is a brief introduction to each network:

- ResNet (O'Shea et al., 2018): The ResNet model is a convolutional neural network with multiple stacked residual blocks. It efficiently extracts important features from time-series data through skip connections and convolution operations, demonstrating strong representational capabilities.

- DAELSTM (Ke & Vikalo, 2021): DAELSTM is a model that combines an LSTM autoencoder and deep fully connected layers, suitable for feature learning and classification of time-series data. It extracts temporal features using the LSTM network, performs data reconstruction with the autoencoder, and classifies using the fully connected layers.

- MCLDNN (Xu et al., 2020): The MCLDNN model integrates Convolutional Neural Networks (CNN) and Long Short-Term Memory networks (LSTM). It extracts local features through convolution layers, captures temporal dependencies with LSTM layers, and classifies using fully connected layers.

- Three-Stream (Liang et al., 2021): The Three-Stream model is designed to handle signal data with multiple input channels. It consists of three main sub-networks (one for each input channel), each processing different input features independently, and finally merging the extracted features for classification.

- FEAT (Chen et al., 2023): FEAT model effectively extracts information and classifies electromagnetic signals by combining time-series data processing, feature extraction, and multi-head attention mechanisms.

### A.5. Evaluation Metrics

We use three metrics to evaluate the model's performance: F1-Score (F1), average model classification accuracy across all signal-to-noise ratios (ACC), and highest model classification accuracy at a single signal-to-noise ratio (H-ACC). F1-score is a commonly used metric in classification tasks as it balances precision and recall, providing a comprehensive assessment of the model's overall performance across different classes. Since a model's classification performance varies across different signal-to-noise ratios, we use ACC and H-ACC to explore the average effect of FR across all signal-to-noise ratios and its effect on a single signal-to-noise ratio, respectively. Assume there are C signal-to-noise ratios in a K-class classification task, where $TP_{ij}$ represents the true positives (TP) for class j at signal-to-noise ratio i, the number of samples where the actual class is j and it is predicted as j; $FP_{ij}$ represents the false positives (FP) for class j at signal-to-noise ratio i, the number of samples where the actual class is not j but predicted as j; $FN_{ij}$ represents the false negatives (FN) for class j at signal-to-noise ratio i, the number of samples where the actual class is j but predicted as a different class. The formulas for the three metrics are as follows:

$$
\begin{cases}
\text{F1} = \frac{1}{K} \sum_{i=1}^{C} \sum_{j=1}^{K} 2 \times \frac{a \times b}{a+b}, \\
\text{ACC} = \sum_{i=1}^{C} \sum_{j=1}^{K} b, \\
\text{H} - \text{ACC} = \max_{i \in C} \{ \sum_{j=1}^{K} b \}.
\end{cases}
\tag{12}
$$

where $a = \frac{TP_{ij}}{TP_{ij}+FP_{ij}}, b = \frac{TP_{ij}}{TP_{ij}+FN_{ij}}$.

