# OpenReview forum: "Robust Automatic Modulation Classification with Fuzzy Regularization"
_ICML.cc/2025/Conference — ICML 2025 spotlightposter_

### Official Review · Reviewer_NJBZ · 2025-02-20

**Overall Recommendation:** 4

**Summary:**

The paper introduces Fuzzy Regularization (FR) as a novel solution to mitigate prediction ambiguity in Automatic Modulation Classification (AMC). This ambiguity is caused by similar characteristics between modulation schemes, especially under noisy conditions. The FR approach is characterized by three key features: modeling prediction ambiguity, dynamic sample reweighting through adaptive loss scaling, and promoting margin maximization between similar modulation classes. The experimental results show that FR significantly improves the robustness and classification accuracy across various datasets and noise levels.

## update after rebuttal

I will keep my score.

**Claims And Evidence:**

The claims about FR improving model performance and robustness are supported by extensive experimental evidence, including results across multiple datasets (RadioML 2016.10a, 2016.10b, and 2018.01A). The improvements in accuracy and robustness are clearly documented. However, the theoretical analysis and description of the FR mechanism could be clearer and more rigorously formalized, particularly concerning the adaptive gradient mechanism.

**Essential References Not Discussed:**

The paper cites key references, but it could benefit from more discussion on the connection to related regularization techniques, particularly those addressing prediction uncertainty or ambiguity, such as label smoothing, entropy regularization, and adversarial robustness techniques.

**Experimental Designs Or Analyses:**

The experimental designs are sound, with clear benchmarks and comparisons with other state-of-the-art (SOTA) methods. The paper includes a comprehensive evaluation across different noise levels, which validates the robustness of the FR method. The use of multiple datasets and the careful selection of evaluation metrics (F1-Score, ACC, H-ACC) further strengthens the experimental analysis.

**Methods And Evaluation Criteria:**

The proposed method, Fuzzy Regularization (FR), is appropriate for addressing the problem of prediction ambiguity in AMC tasks, particularly under noisy conditions. The use of standard benchmark datasets and performance metrics like F1-Score, ACC, and H-ACC is suitable for evaluating the method's effectiveness. The experimental design, which compares FR with existing methods, helps establish the validity of FR as an enhancement to AMC models.

**Other Comments Or Suggestions:**

Some sections could benefit from clearer writing and better organization, especially the presentation of the mathematical formulas and experimental setup. It would also be helpful to provide a more detailed comparison with other ambiguity-handling techniques.

**Other Strengths And Weaknesses:**

The paper is strong in its originality, proposing a novel regularization mechanism that specifically targets prediction ambiguity, a critical issue in AMC. The experimental results are robust and demonstrate the method's efficacy across various noise conditions. However, the explanation of the FR mechanism lacks some clarity, particularly in its formalization. The paper would benefit from a deeper theoretical exploration of the method.

**Questions For Authors:**

- Can you provide a more formal and detailed explanation of the adaptive gradient mechanism used in Fuzzy Regularization? This would help clarify how it dynamically adjusts during training.
- How does the FR mechanism compare to label smoothing or other entropy-based regularization methods in terms of performance and robustness? Would it be beneficial to combine these techniques?

**Relation To Broader Scientific Literature:**

The paper positions itself within the broader context of automatic modulation classification, comparing FR to other regularization techniques and deep learning models. It highlights the gap in the literature regarding the explicit handling of prediction ambiguity and offers a new approach with FR. However, the paper could better compare FR with other regularization strategies specifically designed for handling ambiguity in classification tasks, such as label smoothing or other entropy-based methods.

**Theoretical Claims:**

The theoretical claims around prediction ambiguity and the need for a regularization mechanism to address it are well-grounded. However, the mathematical formulation of the FR mechanism could benefit from more clarity. The paper lacks a formal definition of how the adaptive gradient mechanism works mathematically, which would strengthen the theoretical claims.

---

> ### Author Rebuttal · Authors · 2025-03-30
>
> Thank you for professional comments. We have tried our best to address your questions and revised our paper by following suggestions from all reviewers.
>
> **Q1: Can you provide a more formal and detailed explanation of the adaptive gradient mechanism used in Fuzzy Regularization? This would help clarify how it dynamically adjusts during training.**
>
> RE:Thank you for your question, it helps to articulate our work more clearly. We have explained the establishment of Eq. (6) in more detail and hope that this will help you to understand the automatic gradient mechanism in FR. First choose the first k predicted values of each sample to calculate the degree of fuzziness ($\mathrm{M}$), and the theoretical maximum value $\mathrm{M}\_{\mathrm{max}}=\left(\sum\_{i=1}^{k}\hat{y}\_{i}\right)^{2}-{k\mu}^2$, minimum value $\mathrm{M}\_{\mathrm{min}}=k\times\left(\frac{\sum\_{i=1}^{k}\hat{y}\_{i}}{k}\right)^{2}-{k\mu}^2$, and mean value $\mu=\frac{\sum\_{i=1}^{k}\hat{y}\_{i}}{k}$ for a single sample are known. Then  $\mathrm{M}$ can be expressed as: $\mathrm{M}=\sum\_{i=1}^\mathrm{k}\left(\hat{y}\_{i}-\mu\right)^2=\sum\_{i=1}^\mathrm{k}\hat{y}\_{i}^2-{k\mu}^2$; we focus only on the distribution state, so by normalizing the formula we get $\mathrm{M\_{norm}=\frac{M-M\_{min}}{M\_{max-M\_{min}}}=\frac{\sum\_{i=1}^{k}\hat{y\_{i}}^{2}-k\times(\frac{\sum\_{i=1}^{k}\hat{y\_{i}}}{k})^{2}}{(\sum\_{i=1}^{k}\hat{y\_{i}})^{2}-k\times(\frac{\sum\_{i=1}^{k}\hat{y\_{i}}}{k})^{2}}}$.The automatic gradient mechanism is realized by the log(\*) function to correct the gradient of $M\_{norm}$. The absolute value of the gradient of the log(\*) function decays symmetrically when it deviates from the central axis, which can satisfy the second definition of the design of FR. The final loss of a single sample is: $\mathrm{Loss}=\log(\mathrm{M}\_{\mathrm{norm}})=\log\left(\frac{\sum\_{i=1}^k\hat{y}\_i^2-k\times\left(\frac{\sum\_{i=1}^k\hat{y}\_i}{k}\right)^2}{\left(\sum\_{i=1}^k\hat{y}\_i\right)^2-k\times\left(\frac{\sum\_{i=1}^k\hat{y}\_i}{k}\right)^2}\right)$.
>
> **Q2:How does the FR mechanism compare to label smoothing or other entropy-based regularization methods in terms of performance and robustness? Would it be beneficial to combine these techniques?**
>
> RE:We would like to thank the reviewers for their valuable comments. First we define Ent to denote the entropy-based method, LS to denote the label smoothing method, and FRLS to denote the joint training of the label smoothing and FR methods. NF_* indicates that the noise factor of the dataset is *.For the sake of intuition and brevity, the data recorded is the performance difference between the two methods, e.g. Ent_FR is the value obtained by subtracting the entropy-based model accuracy from the FR model accuracy in the original article.
>
> Through Table I we find that the performance difference of entropy-based methods compared to WF methods is both good and bad. We analyze that the reason for the deterioration of performance may be due to the gradient update of the entropy function and other problems, which is one of the difficulties solved in this paper. Although the performance of the entropy-based method can be close to the FR method in some tasks, the performance is weaker than the FR method in general tasks. This also shows that the FR method is better than the entropy-based method.
>
> Table 1
>
> ||NF_0|NF_0|NF_20%|NF_20%|NF_40%|NF_40%|NF_60%|NF_60%|
> |-|-|-|-|-|-|-|-|-|
> ||Ent_FR|Ent_WF|Ent_FR|Ent_WF|Ent_FR| Ent_WF| Ent_FR| Ent_WF|
> |DAE|-4.18%|-1.05%|-0.83%|+1.6%|-0.83%|+0.9%|-4.77%|-1.3%|
> |FEA|-0.42%|+0.03%|-2.84%|+0.3%|-0.07%|+0.4%|-0.17%|+0.24%|
> |MCL|-3.56%|+1.52%|-0.82%|+2%|-0.4%|+0.65%|-0.14%|+0.5%|
> |Res|-1.51%|-0.18%|-1.05%| -0.1%|-2.34%|-0.6%|-4.13%|-0.1%|
> |Thr|-6.97%|-4.71%|-5.84%|+2.3%|-8.59%|-2.1%|-9.03%|-1.1%|
>
> With Table 2 we find that the FRLS models generally outperform the models trained with label smoothing loss, which further demonstrates the FR validity. Meanwhile, we find that the model trained by FRLS joint loss does outperform the model trained by FR supervision on some tasks. This may be due to the fact that LS optimizes the target labels, which make the values of non-target classes non-zero.This helps the model to learn more information from other classes. FR mainly focuses on the model's predictive distribution information. The two focus on different information, so the performance of the model can be further optimized when trained jointly. We will continue to study the joint training strategy in depth.
>
> Table 2
>
> ||NF_0|NF_0|NF_20%|NF_20%|NF_40%|NF_40%|NF_60%|NF_60%|
> |-|-|-|-|-|-|-|-|-|
> ||FRLS_FR|FRLS_LS|FRLS_FR|FRLS_LS|FRLS_FR|FRLS_LS|FRLS_FR|FRLS_LS|
> |DAE|-0.1%|+1%|+1.9%|+1.3%|+0.9%|+1.7%|+1.9%|+0.7%|
> |FEA|+0.9%|+1.9%|+0.8%|+0.6%|+0.6%|-0.5%|-0.8%|+1.2%|
> |MCL|+1.4%|+0.7%|+1.7%|+2.1%|+1.8%|+0.4%|+1%|+0.2%|
> |Res|-0.8%|+0.3%|+0.8%|+0.6%|+1.2%|+0.7%|-1.9%|+0.4%|
> |Thr|+0.6%|+0.1%|+0.9%|+1.2%|+2.2%|+3.1%|+2%|+2%|

---

### Official Review · Reviewer_X5bz · 2025-03-04

**Overall Recommendation:** 4

**Summary:**

This paper proposes a regularization method aimed at enhancing classification performance in signal classification tasks, particularly for those with low signal-to-noise ratios. It achieves this by constraining the model's predictive ambiguity for samples during the task, thereby increasing the inter-class distance between different categories and reducing the intra-class distance within the same category, which in turn improves classification performance. The authors have validated the effectiveness of this regularization on both synthetic and benchmark datasets.

**Claims And Evidence:**

Yes.

**Essential References Not Discussed:**

This paper includes key relevant literature to help readers understand the research background and significance of the issue.

**Experimental Designs Or Analyses:**

I have reviewed the experimental section, including Experiments Settings (4.1),
Comparison with Other Methods(4.2), Generalizability of the Fuzzy Regularization (FR)(4.3), Robustness of the Fuzzy Regularization (FR)(4.4), Training Behaviour Analysis(4.5), Parameter Sensitivity Analysis(4.6).

**Methods And Evaluation Criteria:**

Yes. The method proposed in this paper is effective in enhancing the performance of classification tasks. The evaluation criteria selected in this paper can effectively distinguish the performance differences between the proposed method and the baseline.

**Other Comments Or Suggestions:**

In the appendix, the numbering for Datasets should be A.3 instead of A.4.

**Other Strengths And Weaknesses:**

Strengths:
1. The paper elaborates on the research motivation in detail, with a well-structured and fluently written narrative.
2. The paper discusses the existing issues in automatic modulation classification tasks and provides corresponding solutions.
3. The experimental results consistently demonstrate that models incorporating FR regularization outperform the baseline, highlighting the effectiveness and versatility of this method in signal classification tasks.

Weaknesses:
1. The work appears to bear some resemblance to curriculum learning. Could the authors provide a detailed explanation of the similarities and differences between these two methods?
2. The backward derivation of this regular derivative is not seen in the text, but it is important for the subsequent optimization of the model, and it is hoped that the author can add this part of the content.

**Questions For Authors:**

See weaknesses.

**Relation To Broader Scientific Literature:**

It contributes to the field of signal classification by introducing a regularization technique that enhances the reliability of classification tasks. This regularization technique is proposed to address the phenomenon of ambiguity, and its effectiveness has been validated through multi-dimensional experiments.

**Theoretical Claims:**

There are no theoretical claims in the paper.

---

> ### Author Rebuttal · Authors · 2025-03-30
>
> Thank you for professional comments. We have tried our best to address your questions and revised our paper by following suggestions from all reviewers.
>
> **W1: The work appears to bear some resemblance to curriculum learning. Could the authors provide a detailed explanation of the similarities and differences between these two methods?**
>
> RE: This is very good question. There are two differences between FR and curriculum learning.  Firstly, the training strategy is different. Curriculum learningis a two-stage learning strategy: learning the easy samples then the difficult samples [1]. FR did not have this strategy and focused on marginal samples(which can be regarded as the difficult samples in course learning) from the beginning.  The second and biggest difference is that the both of them utilize different information. Since FR can measure the degree of predictive ambiguity of the samples, it utilizes more information about the predictive distribution of the samples than course learning.
>
> [1] Liu Y, Wang J, Xiao L, et al. Foregroundness-aware task disentanglement and self-paced curriculum learning for domain adaptive object detection[J]. IEEE Transactions on Neural Networks and Learning Systems, 2023.
>
> **W2: The backward derivation of this regular derivative is not seen in the text, but it is important for the subsequent optimization of the model, and it is hoped that the author can add this part of the content.**
>
> RE: In order to facilitate the detailed explanation of the subsequent derivation process, we provide a more detailed additional description of Eqs. (7)(8), which is supplemented as follows:
> $$\mathrm{F}(\hat{y}\_{j})=\frac{\sigma}{T(\hat{y}\_{j},\tau)\sqrt{2\pi}}exp\left[-\frac{\sigma^2}{2}log\left(T\left(\hat{y}\_{j},\tau\right)\right)^2\right]$$
> $$\mathrm{s.t.}\quad\begin{cases}\sigma=-\frac{1}C\sum\_{j=1}^c\frac{\sum\_{i=1}^{k}\widehat{y}\_{ji}log(\widehat{y}\_{ji})}{log(\tau)}, \\\T(\widehat{y}\_{j},\tau)=\frac{\sum\_{i=1}^{k}\widehat{y}\_{ji}^{2}-k\left(\frac{\sum\_{i=1}^{k}\widehat{y}\_{ji}}{k}\right)^2}{\left(\sum\_{i=1}^{k}\widehat{y}\_{ji}\right)^2-k\left(\frac{\sum\_{i=1}^{k}\widehat{y}\_{ji}}{k}\right)^2}.&&&\end{cases}$$
>
> where $\hat{y}\_{j}$ denotes the jth sample, $\hat{y}\_{ji}$ denotes the probability that the sample belongs to class i, C denotes the sample size of the batch, $\tau$ denotes the $\tau$ classification task and k denotes the selection of the first k values.
>
> 1. Calculating derivatives of intermediate variables
>
> 1.1 The derivative of $\sigma$ with respect to $\widehat{y}\_{ji}$
>
> $\frac{\partial\sigma}{\partial\hat{y}\_{ji}}=-\frac{1}{\mathrm{Clog}\tau}\cdot\frac{\partial}{\partial\hat{y}\_{ji}}(\hat{y}\_{ji}\mathrm{log}\hat{y}\_{ji})=-\frac{\log\hat{y}\_{ji}+1}{C\log\tau}$.
>
> 1.2 The derivative of $T(\widehat{y}\_j,\tau)$ with respect to $\widehat{y}\_{ji}$
>
> Let $S\_1=\sum\_{i=1}^k\hat{y}\_{ji}, S\_2=\sum\_{i=1}^k\hat{y}\_{ji}^2$.
> Then $T=\frac{S\_2-\frac{S\_1^2}{k}}{S\_1^2-\frac{S\_1^2}{k}}=\frac{kS\_2-S\_1^2}{(k-1)S\_1^2}$, we let the numerator in $T$ be N and the denominator D.
>
> So $\frac{\partial T}{\partial\hat{y}\_{ji}}=\frac{\partial N/D}{\partial\hat{y}\_{ji}}=\frac{\partial N}{\partial\hat{y}\_{ji}}\cdot\frac{1}{D}-\frac{N}{D^2}\cdot\frac{\partial D}{\partial\hat{y}\_{ji}}$.
>
> 1.2.1 The derivative of the molecule N
>
> Due $\frac{\partial S\_2}{\partial\hat{y}\_{ji}}=2\hat{y}\_{ji}, \frac{\partial S\_1}{\partial\hat{y}\_{ji}}=1$. So $\frac{\partial N}{\partial\hat{y}\_{ji}}=k\cdot\frac{\partial S\_2}{\partial\hat{y}\_{ji}}-2S\_1\cdot\frac{\partial S\_1}{\partial\hat{y}\_{ji}}=2k\hat{y}\_{ji}-2S\_1$.
>
> 1.2.2 The derivative of the denominator D
>
> $\frac{\partial D}{\partial\hat{y}\_{ji}}=(k-1)\cdot2S\_1\cdot\frac{\partial S\_1}{\partial\hat{y}\_{ji}}=2(k-1)S\_1$.
>
> 1.2.3 The final derivative of $T(\widehat{y}\_{j},\tau)$ with respect to $\widehat{y}\_{ji}$
>
> $\frac{\partial T}{\partial\hat{y}\_{ji}}=\frac{2k\hat{y}\_{ji}-2S\_1}{(k-1)S\_1^2}-\frac{(kS\_2-S\_1^2)\cdot2(k-1)S\_1}{(k-1)^2S\_1^4}=\frac{2k(\hat{y}\_{ji}S\_1-S\_2)}{(k-1)S\_1^3}$.
>
> 2. Chain rule for derivation
>
> Due $\frac{\partial\ln F}{\partial\hat{y}\_{ji}}=\frac{1}{\sigma}\frac{\partial\sigma}{\partial\hat{y}\_{ji}}-\frac{1}{T}\frac{\partial T}{\partial\hat{y}\_{ji}}-\sigma(\log T)^2\frac{\partial\sigma}{\partial\hat{y}\_{ji}}-\frac{\sigma^2\log T}{T}\frac{\partial T}{\partial\hat{y}\_{ji}}$.
>
> So $\frac{\partial F}{\partial\hat{y}\_{ji}}=F\cdot\frac{\partial\ln F}{\partial\hat{y}\_{ji}}=F(\hat{y}\_j)\cdot\left[\frac{\partial\sigma}{\partial\hat{y}\_{ji}}\left(\frac{1}{\sigma}-\sigma(\log T)^2\right)+\frac{\partial T}{\partial\hat{y}\_{ji}}\left(-\frac{1}{T}-\frac{\sigma^2\log T}{T}\right)\right]$.
>
> **Other Comments Or Suggestions**
>
> We will correct the numbering problem in the appendix.

---

### Official Review · Reviewer_F7As · 2025-03-06

**Overall Recommendation:** 4

**Summary:**

This paper proposes a method to improve the reliability of signal classification models by means of fuzzy regularization. Starting from the prediction fuzzy phenomenon, the authors first experimentally prove that the prediction fuzzy phenomenon is a common phenomenon in automatic modulation recognition, and then deeply discuss the impact of this phenomenon on the model performance and propose a corresponding solution, i.e., fuzzy regularization. Finally, the effectiveness and generalization of the method are verified through experiments.

**Claims And Evidence:**

Yes.

**Essential References Not Discussed:**

No significant omissions of relevant literature were detected.

**Experimental Designs Or Analyses:**

I have meticulously reviewed the experimental section in the fourth chapter of the article, and the overall experiments are well-conceived. I have also noted that the authors have ensured consistency across various controllable parameters in the experiments, such as random seeds, which greatly contributes to the fairness of the experiments.

**Methods And Evaluation Criteria:**

Yes.

**Other Comments Or Suggestions:**

C in equation (8) does not find a specific meaning in the context.
It is best to cite reference sources for the dataset.

**Other Strengths And Weaknesses:**

Strengths:
1.The overall content of this paper is complete and clear. The author elaborates on the process of identifying problems, exploring issues, solving problems, and verifying the validity of the methodology.
2. The design of the fuzzy regularity proposed in this paper is not complicated, but the overall design is more skillful and at the same time more comprehensively considered. For example, when designing the fuzzy regularity in Section 3.3, the authors consider eliminating the influence of the model predictive distribution on the predicted fuzzy values through normalization.
3.The validity and generalizability of the method were verified by experiments.

Weaknesses:
1.Does FR sharpen misclassified samples?
2. Has the author considered that FR regularization might lead to overconfidence issues?
3.The parameter k in the regularization proposed by the authors is quite significant, but how should I go about selecting this parameter k?

**Questions For Authors:**

Please See Weaknesses.

**Relation To Broader Scientific Literature:**

The authors have made pivotal contributions to the field of signal classification. They have conducted an in-depth analysis of the ambiguity phenomena in signal classification tasks and proposed an effective solution to address these issues.

**Theoretical Claims:**

There are no theoretical claims.

---

> ### Author Rebuttal · Authors · 2025-03-30
>
> Thank you for professional comments. We have tried our best to address your questions and revised our paper by following suggestions from all reviewers.
>
> **W1: Does FR sharpen misclassified samples?**
>
> RE: Thanks for your valuable comment. We can suppress this phenomenon by adjusting the hyperparameters $\gamma$. In the training process, the model is trained with the guidence of both the FR and the cross-entropy function. When the FR originally misclassifies the sample, its corresponding sample' prediction probability value becomes sharp. Although this leads to the decrease of the FR value, but at the same time, it will also lead to the increase of the value of the cross-entropy function. Hence, we can inhibit the occurrence of this phenomenon by selecting an appropriate $\gamma$ value. That is to say, the decrease of the FR loss is smaller than the increase of the cross-entropy loss  when the wrong prediction becomes sharp.
>
> For the appropriate value, we have pointed out in Section 4.6 of the paper that the proposed model usually performs better when the FR canonical value is two orders of magnitude different from the value of the cross-entropy function.
>
> **W2: Has the author considered that FR regularization might lead to overconfidence issues?**
>
> RE: We thank the reviewer for the insightful comments. The work [1,2] shows that overconfidence mainly targets the following two types of samples. The one is that the model's predicted probability value for the class is very high, but the classification result is wrong. And the other is that the model's predicted probability value for the class is very high, but the actual accuracy is much smaller than the predicted probability value.
>
> Firstly, for the first class of samples, we have explained in W.1 that the generation of this class is suppressed by choosing an appropriate hyperparameter $\gamma$. For the second type of samples, we have introduced an adaptive gradient mechanism in the design of FR. This mechanism ensures that the gradient returned by the FR decreases as the value of the predicted probability increases. As a result, the model's predicted probability value for the sample does not always keep increasing. In summary, FR does not lead to overconfidence issues.
>
> [1] A. Roitberg, et al. Is My Driver Observation Model Overconfident? Input-Guided Calibration Networks for Reliable and Interpretable Confidence Estimates, in IEEE Transactions on Intelligent Transportation Systems, vol. 23, no. 12, pp. 25271-25286
> [2] D Wu, et al. The overconfident and ambiguity-averse newsvendor problem in perishable product decision[J]. Computers & Industrial Engineering, 2020, 148: 106689.
>
> **W3: The parameter k in the regularization proposed by the authors is quite significant, but how should I go about selecting this parameter k?**
>
> RE: For the choice of the value of $k$, it is suggested to set the maximal number of confusion categories. This is because if a deterministic class has $m$ similar classes, then the model's output value for these $m+1$ classes is closest. This is helpful for FR to accurately measure the degree of predictive ambiguity of a sample.
>
> In this paper, $k$ is set the number of coding categories of the modulation with the most coding types in the dataset. For example, in a quadrature dataset, the categories are: 'QAM16', 'QAM64', 'QPSK' and 'WBFM'. Of these, QAM has two coding types, while the other modulations each have only one coding type. Therefore, the value of $k$ is initially chosen to be 2.
>
> **Other Comments Or Suggestions**
>
> C represents the number of samples included in a single round of training for the model.
> We cite the source of the dataset in Text 4.1Experiments Settings and indicate the download link for the dataset in Appendix A.3.

---

### Official Review · Reviewer_GYDk · 2025-03-12

**Overall Recommendation:** 3

**Summary:**

This article primarily introduces a fuzzy regularization method applicable to the field of automatic modulation. The authors first measure the prediction ambiguity by an entropy function or a regular function, and then gradually introduced adaptive gradient and exponential normal distribution to further optimize the metrics to design FR. The authors' main contribution is that FR mitigates ambiguities and improves the classification accuracy in the signal classification task, and the method is more effective at low signal-to-noise ratios.

**Claims And Evidence:**

Yes

**Essential References Not Discussed:**

There are no additional relevant literatures that need to be supplemented.

**Experimental Designs Or Analyses:**

I have checked all the experiments in the experimental section.

**Methods And Evaluation Criteria:**

The methods and evaluation criteria proposed in the paper are effective for addressing the intended problems.

**Other Comments Or Suggestions:**

See Weaknesses

**Other Strengths And Weaknesses:**

Strengths:

1. This paper focuses on discussing an important issue.
2. The methodology presented in this paper is simple, novel, yet effective.
3. The authors have validated the effectiveness of the method through experiments from multiple perspectives.

Weaknesses:

1. Section 4.5 lacks experimental validation regarding the time taken for a single training round. Fast convergence in terms of training epochs does not necessarily equate to rapid convergence in actual time. It is hoped that the authors can provide more detailed experiments in this aspect.
2.  As dataset sizes continue to expand, could this regularization potentially reduce the training efficiency of the model?

**Questions For Authors:**

1. Section 4.5 lacks experimental validation regarding the time taken for a single training round. Fast convergence in terms of training epochs does not necessarily equate to rapid convergence in actual time. It is hoped that the authors can provide more detailed experiments in this aspect.
2.  As dataset sizes continue to expand, could this regularization potentially reduce the training efficiency of the model?

**Relation To Broader Scientific Literature:**

The core contribution of this paper is proposing a novel and effective regularization method for signal classification tasks. The authors elucidate the negative impact of the ambiguity phenomenon on the task and design a corresponding regularization constraint to mitigate this phenomenon, thereby enhancing the performance of the classification task.

**Theoretical Claims:**

The paper does not involve theoretical claims.

---

> ### Author Rebuttal · Authors · 2025-03-30
>
> Thank you for professional comments. We have tried our best to address your questions and revised our paper by following suggestions from all reviewers.
>
> **W1: Section 4.5 lacks experimental validation regarding the time taken for a single training round. Fast convergence in terms of training epochs does not necessarily equate to rapid convergence in actual time. It is hoped that the authors can provide more detailed experiments in this aspect.**
>
> RE: Thanks for your professional suggestion. The reviewer's concern: although FR converges faster in terms of training epochs; However, if FR brings too much time consumption in a single round, then FR cannot be considered as effective in speeding up the experimental convergence. According to this suggestion, we conduct addtional experiment and record the shortest training time, the longest training time, and the average training time per round for all rounds in the training process. During the experiments we ensured the consistency of the controllable parameters. The experimental results are as follows:
>
> |||Noise2016a_20%|||Noise2016a_40%|||Noise2016a_60%||
> |-|:-:|:-:|:-:|:-:|:-:|:-:|:-:|:-:|:-:|
> ||MIN|MAX|AVG|MIN|MAX|AVG| MIN|MAX|AVG|
> |FR|0.75s|2.14s|0.90s|0.76s|1.60s|0.91s|0.75s|1.92s|0.91s|
> |Without_FR|0.71s|1.91s|0.89s|0.77s|1.50s|0.92s|0.70s|1.67s|0.80s|
>
> From the experimental result, it can be seen that there is no significant change in the shortest training time, longest training time, and average training time per round whether the  FR is added or not. Combined with the fact that FR can speed up the convergence rounds during training as mentioned in the manuscript, it can be shown that FR can speed up the convergence in actual time.
>
> **W2: As dataset sizes continue to expand, could this regularization potentially reduce the training efficiency of the model?**
>
> RE: Thanks for the valuable question. The calculation of FR focuses on $\sigma$ and $T(\hat{y}_{i},\tau)$ and their values are obtained by matrix operations. Therefore, this process does not consume too much time. The experiment further proved that FR does not lead to a slowdown in training efficiency. We conducted experimental analyses on three different sized datasets, and recorded the shortest training time, the longest training time, and the average training time per round to the three datasets of different sizes. The specific experimental results are as follows:
>
> |||2016a(611.23MB)|||2016b(3.26GB)|||2018(19.98GB)||
> |-|:-:|:-:|:-:|:-:|:-:|:-:|:-:|:-:|:-:|
> ||MIN|MAX|AVG|MIN|MAX|AVG| MIN|MAX|AVG|
> |FR|0.73s|1.57s|0.85s|3.24s|4.64s|3.59s|93.15s|94.04s|93.42s|
> |Without_FR|0.65s|1.57s|0.77s|2.83s|3.98s|3.15s|92.37s|93.19s|92.55s|
>
> The above experimental results show that the average rounds of training time increases by 0.1s and 0.4s on the smaller datasets Data2016a and 2016b, respectively. It is noted that the average time increase is not more than 1s on Data2018a with  20GB, and  the increase of the training time for the 200 rounds  is not more than three minutes.
> While the performance is improved by three percentage points. This is acceptable in the field of signal recognition.

---

### Decision · Program_Chairs · 2025-05-01

**Decision:**

Accept (spotlight poster)

**Comment:**

This paper proposes a novel framework, namely Fuzzy Regularization-enhanced AMC (FR-AMC),  which integrates uncertainty quantification into the classification pipeline. All reviewers give positive scores. The method presented is simple, novel, yet effective. Extensive experiments verify the effectiveness and generalization ability of this method.